# On the origin of low-valent uranium oxidation state

C. L. Silva [1,2], L. Amidani [1,2] ✉, M. Retegan [3], S. Weiss[2], E. F. Bazarkina[1,2], T. Graubner[4], F. Kraus [4] & K. O. Kvashnina [1,2] ✉

The significant interest in actinide bonding has recently focused on novel compounds with exotic oxidation states. However, the difficulty in obtaining relevant high-quality experimental data, particularly for low-valent actinide compounds, prevents a deeper understanding of 5f systems. Here we show X-ray absorption near-edge structure (XANES) measurements in the high-energy resolution fluorescence detection (HERFD) mode at the uranium $M_4$ edge for the $U^{III}$ and $U^{IV}$ halides, namely $UX_3$ and $UX_4$ (X = F, Cl, Br, I). The spectral shapes of these two series exhibit clear differences, which we explain using electronic structure calculations of the 3d-4f resonant inelastic X-ray scattering (RIXS) process. To understand the changes observed, we implemented crystal field models with ab initio derived parameters and investigated the effect of reducing different contributions to the electron-electron interactions involved in the RIXS process. Our analysis shows that the electron-electron interactions weaken as the ligand changes from I to F, indicative of a decrease in ionicity both along and between the $UX_3$ and $UX_4$ halide series.

Actinide chemistry is richly complex, with the partially filled 5f shell resulting in fascinating and intricate chemical behavior. Additionally, the actinide elements are radioactive, which makes experimental studies more challenging compared to those on non-radioactive materials. In recent years, significant progress has been made in actinide synthetic chemistry, leading to the discovery of several new actinide complexes with exotic oxidation states. These have been identified in elements such as thorium[1], uranium[2–5], neptunium[6], plutonium[7,8], berkelium[9], californium[10], and einsteinium[11]. In particular, the identification of low-valent actinide compounds has enriched the field and highlighted the need for a deeper understanding of the chemical bond and the specific role of 5f electrons.

Addressing this scientific challenge requires reliable experimental techniques to accurately determine the oxidation states and assess the contributions of specific orbitals to chemical bonding. Oxidation state estimations often rely on indirect methods, such as X-ray diffraction and UV–Vis spectroscopy, which have been used for compounds such as $U^I$, $U^{II}$, and $U^{III}$ [2,12,13]. Electron paramagnetic resonance (EPR) is an alternative technique and has been used to detect the trivalent oxidation state of U[14]. It is also sensitive to the pentavalent state of uranium but remains unresponsive to the tetravalent and hexavalent states. X-ray absorption near-edge structure (XANES) in high-energy resolution fluorescence detection (HERFD) mode combined with resonant inelastic X-ray scattering (RIXS) at the $M_{4,5}$ edges is a direct approach to probe oxidation states[15–18]. The shifts in the energy position of the HERFD-XANES main edge allow for highly sensitive determination of oxidation states due to the combination of good spectral resolution and element selectivity of inner-shell spectroscopy. These advanced techniques are typically conducted at large-scale synchrotron facilities[19]. The HERFD-XANES method provides improved energy resolution over conventional XANES by integrating only the maximum characteristic fluorescence emitted by the sample. An X-ray emission spectrometer[20] analyses the emitted X-rays, selecting the maximum intensity of the emission line with an energy resolution of ~1 eV. This

[1]The Rossendorf Beamline at ESRF—The European Synchrotron, CS40220, 38043 Grenoble Cedex, France. [2]Helmholtz-Zentrum Dresden-Rossendorf (HZDR), Institute of Resource Ecology, 01314 Dresden, Germany. [3]European Synchrotron Radiation Facility (ESRF), CS40220, 38043 Grenoble Cedex, France. [4]Fachbereich Chemie, Philipps-Universität Marburg, Hans-Meerwein-Str. 4, 35032 Marburg, Germany. ✉e-mail: lucia.amidani@esrf.fr; kristina.kvashnina@esrf.fr

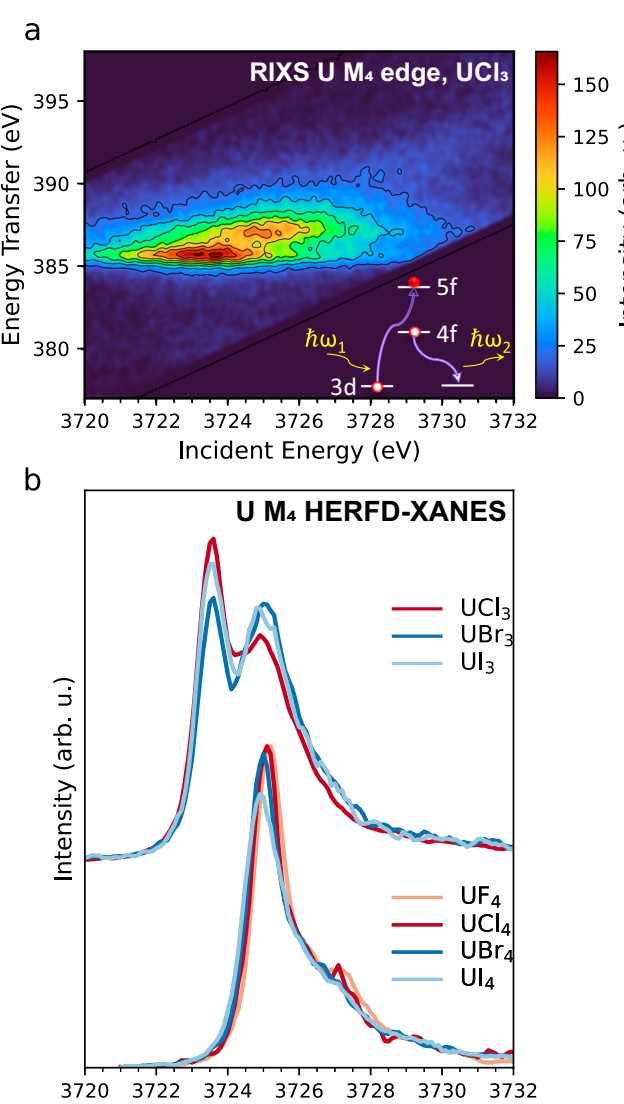

**Fig. 1 | Resonant inelastic X-ray scattering (RIXS) and X-ray absorption near-edge structure in high-energy resolution fluorescence detection mode (HERFD-XANES) measurements at the U $M_4$ edge. a** RIXS of $UCl_3$ displayed as a contour map with axes corresponding to the incident and transferred energies over the U $M_4$ absorption edge and U $M\beta$ emission line. The schematic of the 3d-4f RIXS illustrates the two-step process. In the first step, a 3d electron is promoted to an empty orbital following the absorption of an incoming X-ray photon $\hbar\omega_1$. In the second step, the 3d core-hole is filled by a 4f electron, and the energy is released as an emitted X-ray photon $\hbar\omega_2$. **b** HERFD-XANES data of the $UX_3$ (top) and $UX_4$ (bottom) halides (X = F, Cl, Br, I). The U atom in $UX_4$ has a $5f^2$ ground state configuration, while in $UX_3$ it has a $5f^3$ configuration.

enhanced resolution is particularly significant at the $M_{4,5}$ edges of actinides, making HERFD-XANES a fundamental tool in actinide science[15–18,21–24] since its first application[15]. The shifts in the energy position of the HERFD-XANES main edge allow for highly sensitive determination of oxidation states due to the combination of good spectral resolution and element selectivity of inner-shell spectroscopy. Additionally, the shape of the HERFD-XANES spectrum provides valuable information about the local structure near the actinide atom, which can be further interpreted with the help of electronic structure calculations.

It should be noted that the sensitivity of the HERFD-XANES and RIXS at the $M_{4,5}$ edges to the nature of the actinide bond, particularly to covalency, remains an open question and a topic of intense research

in the field of actinide chemistry. While ligand K-edge XANES can directly measure the ligand–metal mixing[25–27], identifying the effects of bond formation and charge-density redistribution at the $M_{4,5}$ edges is more challenging due to the lack of systematic studies. HERFD-XANES method at the U $M_4$ edge, which probes the 5f states, has been used to determine the $U^{IV}$, $U^V$, and $U^{VI}$ oxidation states of various uranium compounds[15–17]. However, there is a lack of experimental data for low-valent compounds. To the best of our knowledge, no HERFD-XANES data at the $M_4$ edge have been reported for oxidation states below $U^{IV}$. Therefore, the sensitivity of this method to low valence states has yet to be established. Uranium halide systems are ideal to determine the sensitivity of HERFD-XANES to low oxidation states and to investigate the role of electron correlations and of strongly electronegative ligands. They are frequently used in fundamental research to understand the nature of actinide bonding[28–34].

Electronic structure calculations are helpful in interpreting HERFD-XANES and RIXS data and extracting detailed information about the electronic transitions and the nature of the chemical bond[35–51]. The RIXS process is described by the Kramers-Heisenberg equation, which can be calculated using different computational approaches. In the field of RIXS on actinide systems, the most used framework has been the ligand field multiplet theory which allows to include atomic, crystal field, and charge transfer effects[35,43,47,48,50]. Recently, there has been an interest in applying relativistic multiconfigurational approaches to calculate HERFD-XANES[49,51,52] and RIXS[46] on actinides. Moreover, it has been shown that, under certain simplifying assumptions, RIXS can be understood as a convolution of the occupied and unoccupied density of states[16,37,41].

In this work, we show HERFD-XANES spectra at the U $M_4$ edge for the families of $U^{III}$ and $U^{IV}$ halides, namely $UX_3$ and $UX_4$ (X = F, Cl, Br, I). Experimental data on $UX_3$ and $UX_4$ is analyzed with the help of crystal field multiplet calculations, where the electron-electron correlations between the 5f electrons, the 3d, and the 4f core-holes involved in the process are taken into account. For low-valent uranium systems, electron-electron correlations are expected to be stronger than interactions with neighboring atoms, which defined our choice of theoretical approach. Altogether, this study reports the first $M_4$ edge HERFD-XANES and RIXS recorded on $U^{III}$ compounds, confirming the method's sensitivity to low-valent oxidation state, and provides a detailed analysis of the role of electron-electron interactions and ligand effects in shaping the 5f electronic structure for the $UX_3$ and $UX_4$ halides series.

## Results

Figure 1 shows the experimental HERFD-XANES data for the families of $UX_3$ and $UX_4$ halides and the RIXS data of $UCl_3$ as a contour map in a plane of incident energy and energy transfer. The latter represents the energy difference between the incident and the emitted photon energies. Color variations in the plot are due to different scattering intensities. As shown schematically in Fig. 1, RIXS is a two-step process. In the first step, an electron is promoted to an empty orbital following the absorption of an incident photon. At the U $M_4$ edge, this is given by an excitation of a $3d_{3/2}$ electron to the $5f_{5/2}$ shell. In the second step, the core-hole is filled by a core electron, and the energy is released as an emitted X-ray photon. At the U $M_4$ edge, this is given by a $4f_{5/2}$ electron (M$\beta$ emission line). Experimentally, the RIXS process is measured by scanning the incident energy across the $M_4$ absorption edge, and for each incident energy, the characteristic emission line is recorded with the X-ray emission spectrometer. The shape of the U $M_4$ RIXS arises from the allowed electronic transitions between the manifolds of multi-electronic states originating from the electronic configurations of the ground, intermediate, and final states.

The HERFD-XANES spectra correspond to the diagonal cut in the RIXS map taken at the maximum of the M$\beta$ ($4f_{5/2}$–$3d_{3/2}$) emission line. Figure 1b shows the recorded HERFD-XANES data for the $UX_3$ and $UX_4$ sets, revealing significant differences. The III oxidation state of U in $UX_3$

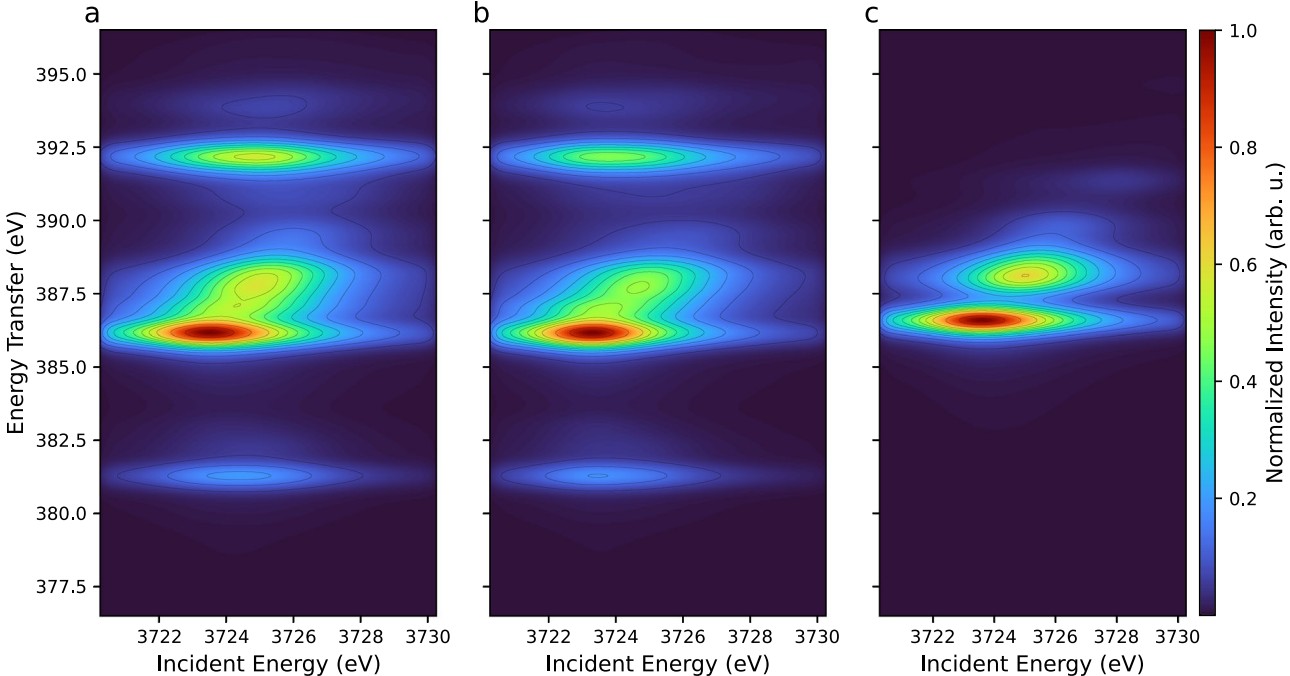

**Fig. 2 | Atomic calculated RIXS at U M$_4$ edge for 5f$^3$ ground state configuration. a** All Slater-Condon integrals are reduced to 80% of their atomic value. **b** Intermediate-state electron-electron interactions are neglected. **c** Final-state electron-electron interactions are neglected.

results in a ~1.6 eV shift of the white line to a lower energy compared to U$^{IV}$. This represents the largest white line shift at the U M$_4$ edge for a unitary change of the oxidation state. In the U$^{IV}$-U$^{V}$-U$^{VI}$ series, the white line shifts are in the order of ~1 eV[15,17]. The shapes of the U$^{III}$ and U$^{IV}$ spectra are also significantly different. While UCl$_4$ (Fig. 1b, bottom) shows only one main peak at ~3725.15 eV, UCl$_3$ (Fig. 1b, top) exhibits a splitting in the white line at ~3723.55 eV and ~3725 eV. Such splitting can be clearly seen in the RIXS plane of Fig. 1a, where two well-separated resonances are observed along the diagonal. As will be shown, the same splitting is well-reproduced in the calculated RIXS for U$^{III}$ by considering the atomic interactions of a 5f$^3$ configuration.

Changes are also present within each halide series. Along with a double profile, UX$_3$ HERFD-XANES spectra show intensity redistribution among the two features depending on the halide ligand. Changes in the UX$_4$ set (Fig. 1b, bottom) are less pronounced but still present. Small shifts in the HERFD-XANES maximum were observed together with differences in intensity and shape of the white line induced by the halide ligands. The trends of the UX$_4$ HERFD-XANES spectra through the halide series can be summarized as follows: there is a lowering in intensity of the main peak and a decrease of the shoulder at ~3727 eV going from F to I, accompanied by an energy shift to lower energies, with the HERFD-XANES maximums of UF$_4$ (3725.2 eV) and UI$_4$ (3724.9 eV) differing by ~0.3 eV.

To understand the spectral changes observed between UX$_3$ and UX$_4$ and within each series, we performed electronic structure calculations. The choice of the method of calculation depends on which approximations are more appropriate for the system under study. In our case, an additional electron is excited into the already partially filled and localized 5f-shell, which means that electron correlations within the 5f-shell and with the core-hole will be strong. As there are strong correlations, a description considering the multiple effects is a good approach[53]. Here, we use crystal field multiplet calculations as implemented in the Quanty code[54]. In the model, we consider the atomic picture with a single U atom and the number of electrons in the relevant shells of the spectroscopic process.

The full RIXS process was calculated, and the extraction of the HERFD-XANES was made through the diagonal cut corresponding to

the maximum of the emission line. The core-to-core 3d-4f RIXS maps are calculated considering the transitions between the 5f$^n$ → 3d$^9$5f$^{n+1}$ → 4f$^{13}$5f$^{n+1}$ electronic configurations, with $n = 2$ for U$^{IV}$ and $n = 3$ for U$^{III}$. The interactions among 5f electrons and between 5f electrons and the 3d and 4f core-holes are described in terms of the Slater-Condon integrals (F$^k$, G$^k$). In the RIXS calculations, the spin-orbit ($\zeta$), the radial Coulomb integrals (F$^k$), the exchange integrals (G$^k$), and the applied crystal field define the multiplets of the ground state 5f$^n$, intermediate-state 3d$^9$5f$^{n+1}$, and final-state 4f$^{13}$5f$^{n+1}$ configurations, as detailed in ref. 43. The values of the Slater-Condon integrals and spin-orbit coupling constants used in the calculations are presented in Supplementary Table 1.

To fundamentally understand the role of specific electronic interactions involved in the spectroscopic process and the origin of the observed features in the RIXS map, we performed an investigation in which we neglected the contributions of the electron-electron interactions in the intermediate and final states of the RIXS process (i.e., the Slater-Condon integrals) in turn. Results for the U$^{III}$ ion are shown in Fig. 2, while Supplementary Table 2 summarizes the ab initio Hartree–Fock values and the reduction factors of the Slater-Condon integrals used in this exercise. Figure 2a shows the theoretical RIXS map of U$^{III}$ when all electron-electron interactions are considered and reduced at 80% of their atomic value, which is the standard procedure when computing ions embedded in a compound. Figure 2b, c show the theoretical RIXS maps when all electron-electron interactions (among 5f and between 5f and the core-hole) are switched off in the intermediate (Fig. 2b) and final (Fig. 2c) states of the spectroscopic process. For all cases, the RIXS main feature has a double profile (transitions at ~386 eV and ~388 eV on the energy transfer scale), which results in the splitting of the white line for the diagonal cut, giving the HERFD-XANES. This clarifies that the presence of two peaks in the U$^{III}$ systems is a fundamental characteristic of a 5f$^3$ configuration as probed by the M$_4$ RIXS process.

In addition to the main features, two other transitions can be seen below (~381 eV) and above (~392 eV) for the atomic calculations (Fig. 2a). When the intermediate-state interactions are neglected (Fig. 2b), these transitions remain nearly in the same position. On the

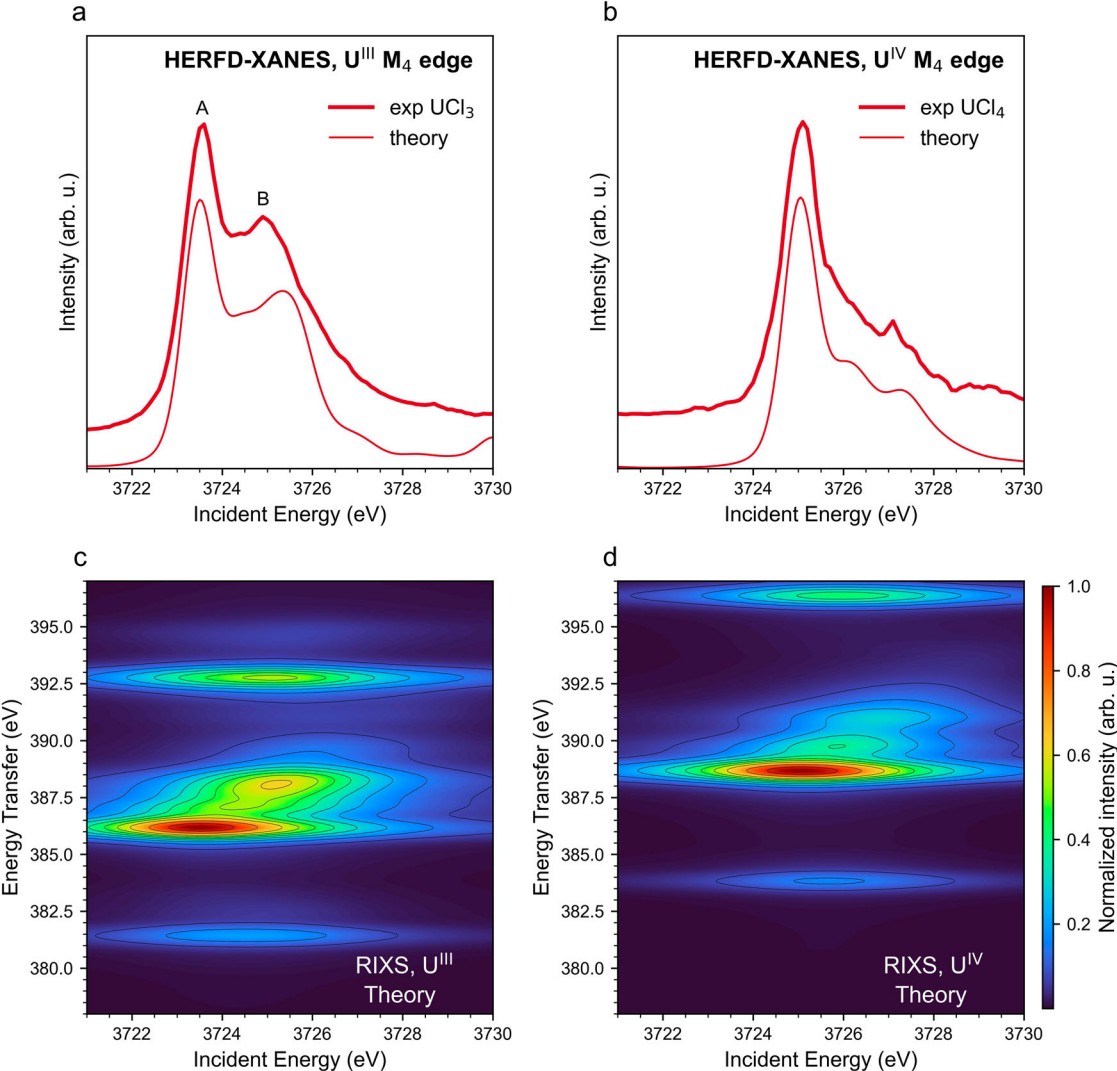

**Fig. 3 | HERFD-XANES and RIXS spectra of UCl₃ and UCl₄.** Experimental HERFD-XANES of **a** UCl₃ and **b** UCl₄ in comparison with calculated 3d-4f RIXS maps for **c** U$^{III}$ and **d** U$^{IV}$. RIXS maps were calculated by crystal field multiplet calculations and the Slater integrals were reduced to 85% and 70% of their atomic values for UCl₃ and UCl₄, respectively. Theoretical HERFD-XANES spectra were obtained by cutting along the diagonal that intersects the maximum intensity of the RIXS maps.

other hand, when the final-state interactions are turned off (Fig. 2c), they overlap with the main feature in the RIXS map. Therefore, the final-state electronic configuration in the RIXS process plays a very important role in the ∼381 eV and ∼392 eV transitions of the standard RIXS map of U$^{III}$ (Fig. 2a). More specifically, such transitions mostly depend on the values of the exchange integrals $G^k(4f,5f)$ and are due to the interactions between the 4f core-hole and the 5f electrons, as previously observed in ref. 43. Indeed, if only the Coulomb integrals $F^k(4f,5f)$ are neglected (see Supplementary Fig. 1), the satellite features are still present in the RIXS map. Similar results are observed for U$^{IV}$ compounds (5f² ground state), where additional features overlap with the main features when final-state electron-electron interactions are neglected. Simultaneously, the intensity and shape of the main feature change (see Supplementary Fig. 2).

In order to reproduce the experimental changes observed and understand the fundamental processes underlying the spectral features, we calculate the atomic RIXS maps for 5f³ and 5f² electronic configurations and extract the theoretical U$^{III}$ and U$^{IV}$ M₄ HERFD-XANES spectra by cutting along the diagonal crossing of the maximum of the RIXS maps. In this approach, the Slater-Condon integrals are scaled to 80% of their atomic values to account for the reduced strength of electron-electron interactions in the compounds. The scaling factor is

further tuned to improve the agreement with the data, with optimal values found between 70% and 85%. Figure 3 shows the best agreement obtained for UCl₃ and UCl₄. The Slater-Condon integrals were scaled to 85% and 76% of their atomic values for UCl₃ and UCl₄, respectively. The reduction factor (e.g., 85% in UCl₃) was kept constant at 85% for UCl₃ and 76% for UCl₄ for the initial, intermediate, and final states of the RIXS process. Figure 3a, b show the experimental HERFD-XANES compared to the theoretical one obtained by cutting the calculated RIXS of Fig. 3c, d, respectively. These two examples of HERFD-XANES and RIXS data recorded on U–Cl systems highlight the spectral differences when the ligand remains the same and the oxidation state is changed. Trivalent and tetravalent U differ significantly from one another due to the additional 5f electron that makes U$^{III}$ have a richer multiplet structure than U$^{IV}$. In particular, U$^{III}$ exhibits a double peak profile (features A and B in Fig. 3a), while U$^{IV}$ has a single main feature followed by a shoulder where two bumps can be distinguished for UCl₄ (Fig. 3b). Our atomic multiplet calculations reproduce these characteristic line shapes very well, indicating that it is mainly the electron-electron interactions that determine the shape of the HERFD-XANES spectra[48].

To determine whether the spectral changes observed within the U$^{III}$ and U$^{IV}$ series correlate with the change in the local environment,

**Table 1 | Local coordination of U atoms, average bond lengths, and space groups for the set of UX$_4$ halides**

| Compound | UF$_4$[65] | UCl$_4$[17] | UBr$_4$[66] | UI$_4$[67] |
|---|---|---|---|---|
| Avg bond length (U–X) | 2.29 Å | 2.77 Å | 2.86 Å | 3.04 Å |
| Coordination number (CN) | 8 | 8 | 7 | 6 |
| Point group | C2 | D4h | Cs | C2 |
| Space group | C2/c | I4$_1$/amd | C2/m | C2/c |

**Table 2 | Local coordination of U atoms, average bond lengths, and space groups for the set of UX$_3$ halides**

| Compound | UCl$_3$[68] | UBr$_3$[69] | UI$_3$[70] |
|---|---|---|---|
| Avg bond length (U–X) | 2.93 Å | 3.08 Å | 3.28 Å |
| Coordination number (CN) | 9 | 9 | 8 |
| Point Group | C3h | C3h | C2v |
| Space group | P6$_3$/m | P6$_3$/m | Cmcm |

we performed multiple calculations using a crystal field model. In this approach, the effect of the local environment is approximated by a static potential perturbing the spherical symmetry of the ion and lifting the degeneracy of the 5f orbitals. The local structure of the U$^{IV}$ and U$^{III}$ systems are given in Tables 1 and 2, respectively. To model the crystal field as accurately as possible, the crystal field parameters describing the splitting of the 5f orbitals were extracted from ab initio calculations. The band structure of each halide system was calculated using Density Functional Theory (DFT) as implemented in the Full Potential Local-Orbital (FPLO) code[55]. The band structure was then projected onto localized Wannier orbitals and the Hamiltonian was diagonalized to extract the parameters for implementing the crystal field. Further details about the computation method can be found in the Supplementary Information. Surprisingly, if the Slater-Condon integrals are kept at the same value for all compounds, the addition of the crystal field term does not significantly change the shape of the HERFD-XANES spectra (see Supplementary Fig. 3c). The comparison between the atomic and crystal field models for the UX$_3$ and UX$_4$ systems is shown in Supplementary Figs. 4 and 5, and the splitting of the 5f orbitals due to the crystal field is presented in Supplementary Fig. 6.

Moreover, to understand the effect of the coordination number (CN) and local geometry on the observed changes of the UX$_4$ systems, we made a simple computational test by calculating the U$^{IV}$ M$_4$ edge HERFD-XANES for UCl$_4$ and UBr$_4$ structures where Cl and Br were substituted with F. In this way, we obtained U coordinated with 8 and 7 fluorine ligands. The results (see Supplementary Fig. 7) show that the spectra for F in the structure of UF$_4$, UCl$_4$, and UBr$_4$ are almost identical, which rules out the significance of the coordination number as a relevant parameter.

The best agreement between data and calculations obtained for the UX$_4$ series (X = F, Cl, Br) are presented in Fig. 4. Figure 4a compares each calculated spectrum to the experimental one to show the excellent agreement reached for each sample. In Fig. 4b, the UX$_4$ set of experimental spectra is shown superimposed and compared with the set of calculations to highlight the similarity among the trends observed in the experiment and theory. In order to have an estimation of the agreement, in Fig. 4b the spectral differences relative to UF$_4$ are shown at the bottom of each set. The calculations reproduce well the trends observed in the experiment. In particular, the shifts of the HERFD-XANES maximum to lower energies, the intensity variations of the white line, and the shoulder at ~3727 eV.

Starting from the values that well reproduced the spectrum of UF$_4$, i.e., 70%, the Slater-Condon integrals were increased to 76%, 80%, and 85% to match the UCl$_4$, UBr$_4$, and UI$_4$, respectively, for the initial,

intermediate, and final states. It should be noted that no shift and re-normalization were applied to the calculated spectra, meaning that the spectral differences are purely induced by the change in parameter values. The reduction of the Slater-Condon integrals from the common value of 80% is usually interpreted as an indication of bond covalency. This is because covalency increases the delocalization of bonding electrons, resulting in the reduced strength of their mutual interactions.

To evaluate whether the spectral trends in the set of U$^{IV}$ systems can be correlated to a significant change in covalency/ionicity as indicated by the 15% variation of the Slater-Condon integrals, we carried out Quantum Theory of Atoms in Molecules (QTAIM) analysis, an electron density topological analysis proposed by Bader[56]. The magnitude of the electron density at the bond critical points (ρBCP) and its Laplacian ($\nabla^2\rho$) at the bond critical point between the U atom and the halogen ligand, as calculated using QTAIM, are shown in Table 3. When BCPs are employed to characterize the nature of a chemical interaction, ρBCP < 0.2 a.u. with $\nabla^2\rho > 0$ indicates a non-covalent interaction[57].

The spectral trends observed for the UX$_4$ set, mainly related to the intensity variation and the position of the main HERFD-XANES peak, correlate with the values of the ρBCP. Such results show that the halide bonds are characterized by ionic interactions (ρBCP < 0.2 a.u.), with the degree of ionic character decreasing from UI$_4$ (ρBCP = 0.048 a.u.) to UF$_4$ (ρBCP = 0.076 a.u.). We therefore interpret the lower values of the Slater-Condon integrals for UF$_4$ as evidence that fluorine ligands are more effective in perturbing the electronic structure of U, with the highest increase in the degree of delocalization of the 5f electrons when compared to the other halides. Although ρBCP values show that all U–X bonds are intrinsically ionic, the observed trends inside the ionic series show that the U-F bond has the highest covalent character, in agreement with previous theoretical studies on UX$_6$ halide complexes[30,32,33]. It is worth noting that according to theoretical studies on U chlorides, the bonding character is strongly dependent on the oxidation state of the metal, with greater ionicity found in low-valent compounds[31]. Therefore, one expects an increased localization of 5f orbitals as the U oxidation state is lowered from IV to III. This was tested by QTAIM analysis (Table 3), which shows a decrease in the ρBCP values as the oxidation state decreases, confirming that U$^{III}$ systems have a higher ionicity than U$^{IV}$ systems.

The case of the UX$_3$ (X = Cl, Br, I) set is very peculiar. Figure 5 shows the best agreement between experiment (Fig. 5a) and theory (Fig. 5b) for the U$^{III}$ systems, and Table 2 provides their local structure. The following discussion will consider only the UCl$_3$ and UBr$_3$ compounds, given that they are isostructural. The experimental spectra of the UX$_3$ set have a double profile, with the intensity of the features varying depending on the halide ligand, as previously discussed in Fig. 1b. Considering the lack of experimental data on U$^{III}$ compounds and the strong tendency of trivalent U to oxidize, the reliability of the measurements and the possibility of beam induced oxidation on the spectra should be examined. The position of the second peak of the U$^{III}$ spectrum corresponds to the energy position of the white line of U$^{IV}$ data (see Fig. 1a). Although a small effect of radiation damage cannot be totally excluded, the spectral differences between UCl$_3$ and UBr$_3$ are beyond radiation damage for the following reasons: the measurements were performed twice under cryogenic conditions, with two sets of different samples, and the same results were obtained. The peak intensities remain the same for data collected on UBr$_3$ using different configurations of the spectrometer (see Supplementary Fig. 8). Our calculations already proved that the double peak profile of U$^{III}$ is characteristic of the 5f$^3$ configuration, a behavior that was previously observed in Pu$^{III}$ compounds (see Supplementary Fig. 9b) where the second HERFD-XANES feature in the Pu$^{III}$ system is also situated at the same energy as the Pu$^{IV}$ peak and was proven to have a different origin[24]. It is worth noting that Pu$^{III}$ is more stable than U$^{III}$, ensuring

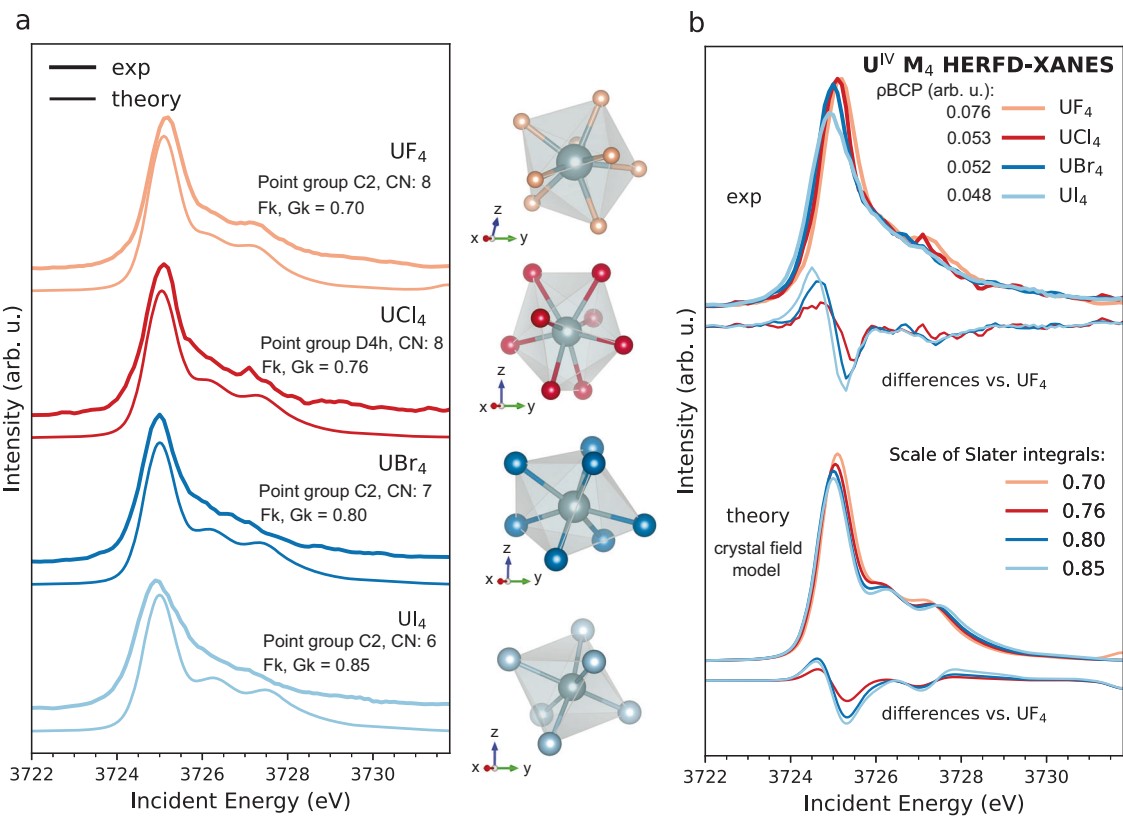

**Fig. 4 | Experimental *versus* calculated HERFD-XANES at the U M₄ edge for the UX₄ halides.** The local coordination of U in each system is shown. **a** Each data is shown compared to the corresponding best simulation. **b** Set of experimental spectra is superimposed and compared with the set of calculations. All differences were calculated compared to the UF₄ spectrum. Theoretical HERFD-XANES spectra were extracted by cutting along the diagonal that intersects the maximum intensity of the RIXS maps, which were obtained by crystal field multiplet calculations. The values of the electron density at the bond critical points (ρBCP) for each system were calculated using the Critic2 code[62] (see Supplementary Methods).

**Table 3 | QTAIM results. Bond critical point (ρBCP) and Laplacian ($\nabla^2\rho$)**

| System | ρBCP (a.u.) | $\nabla^2\rho$ (a.u.) |
|---|---|---|
| UF₄ | 0.076 | 0.293 |
| UCl₄ | 0.053 | 0.114 |
| UBr₄ | 0.052 | 0.092 |
| UI₄ | 0.048 | 0.061 |
| UCl₃ | 0.034 | 0.091 |
| UBr₃ | 0.030 | 0.068 |
| UI₃ | 0.029 | 0.048 |

that the double profile is not correlated with beam damage. Therefore, the aforementioned arguments allow us to claim that the differences observed in the Uᴵᴵᴵ measured spectra are beyond beam damage effects.

To reproduce the differences observed between UCl₃ and UBr₃, we first used the same approach that successfully reproduced the UX₄ series. However, tuning the Slater-Condon integrals using the same reduction factor for the initial, intermediate, and final-state of the RIXS process did not reproduce the effect of intensity variations of the feature B (see Supplementary Fig. 10). Therefore, the effect of reducing different contributions to the electron-electron interactions involved in the RIXS process was investigated to understand the spectral changes. It was found that strong variations in the final-state interactions provide good agreement with experimental data. While keeping all Slater-Condon integrals at 85% of their atomic values resulted in

good agreement with the experimental spectrum for UCl₃, the calculation that best reproduced the data for UBr₃ was the one in which the contributions to the electron-electron interactions involved in the final-state of the RIXS process were kept to 30% of their atomic value (see Fig. 5b). These results suggest that the interaction of the 4f core-hole and the 5f electrons in the final-state, described by the exchange integrals, is strongly affected by the halide ligand in the UX₃ series. The halide ligand would be more effective in screening the 4f core-hole in the case of UBr₃ compared to the case of UCl₃. As the data presented here is the first experimental HERFD-XANES at the M₄ edge of Uᴵᴵᴵ, further investigation is needed to clarify if the pronounced effect of the ligand on the final-state electron-correlations is specific to halides or more general.

## Discussion
The combined experimental and theoretical approach proves the extreme sensitivity of HERFD-XANES and RIXS to probe low oxidation states in complex uranium-containing compounds. The Uᴵᴵᴵ spectra are characterized by a double profile, while Uᴵⱽ spectra have one white line resonance. Compared to Uᴵⱽ species, Uᴵᴵᴵ spectra are shifted to lower energies by -1.6 eV, the highest edge-shift registered for U systems at the M₄ edge. In low-valent uranium compounds, the 5f electronic structure is not expected to contribute substantially to the bonding, and one would rather expect the M₄ edge HERFD-XANES to be insensitive to the system-specific stereochemistry, in analogy to the M₄,₅ edge XANES of trivalent lanthanides. Contrary to expectations, spectral changes are observed within each series. The UX₄ and UX₃ data reveal a degree of 5f sensitivity to the specific local environment. The experimental changes are well-reproduced by calculations of HERFD-

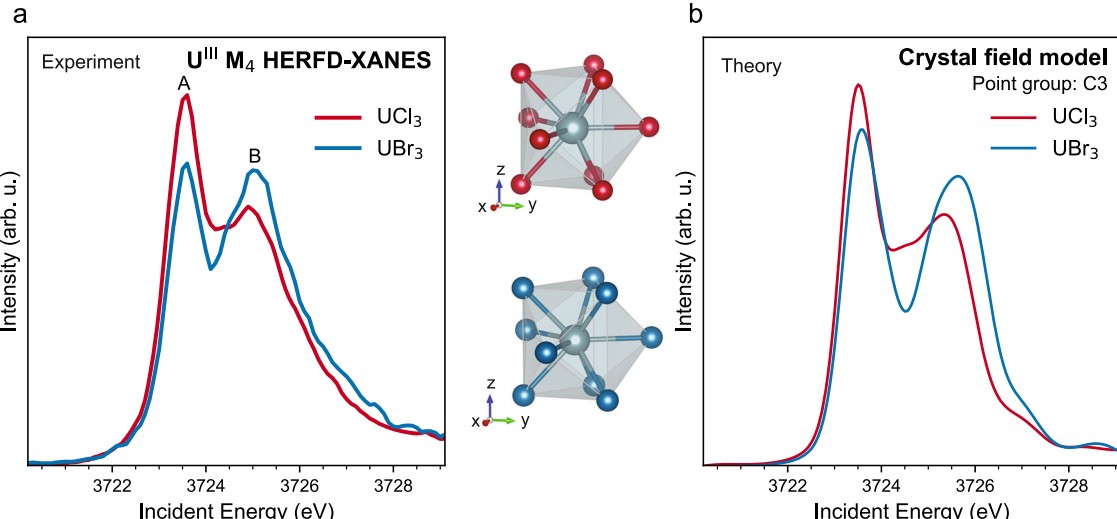

**Fig. 5 | Experimental versus calculated HERFD-XANES at the U M$_4$ edge for the UCl$_3$ and UBr$_3$ compounds.** The local coordination of U in both systems is shown. **a** Experiment and **b** Theory. For UCl$_3$, all Slater-Condon integrals were reduced to 85% of their atomic value. For UBr$_3$, the reduction factor of the initial and intermediate-state interactions was 85%, while the Slater-Condon integrals related to the final-state were scaled down to 30% of their atomic value.

XANES in the framework of crystal field multiplet theory, where the crystal field parameters were extracted ab initio and the Slater-Condon integrals reduced to 70%–85% of their atomic value. Interestingly, the results of the calculations highlight that the trends observed are mostly driven by the change in electron-electron interactions rather than by the lift of the 5f degeneracy induced by the crystal field. In other words, in low-valent U compounds the 5f electrons feel the local environment because it modulates their degree of localization.

This observation raises many questions regarding newly synthesized U$^{II}$ and U$^{I}$ systems. The influence of ligands on the distribution of electron density might have a significant impact also on U$^{II}$ and U$^{I}$ compounds. To address this hypothesis, there is a need to probe electron transitions and electron densities using experimental methods in high-energy resolution mode. It would also be beneficial to simultaneously examine the actinide ion and ligand ions and to extensively investigate the electronic structure of the ligands.

## Methods

### Experimental characterization

M$_4$ edge RIXS and HERFD-XANES spectra were collected at the beamline BM20 (The Rossendorf Beamline) of the European Synchrotron Radiation Facility (ESRF)[58]. Samples were sealed under anoxic conditions in a dedicated cell with 12-micron Kapton film and transported to the beamline under cryogenic conditions. Measurements were performed under a cryostream cooler (Oxford Cryosystem 700). The incident beam energy was selected using a Si(111) double crystal monochromator, and the incident beam was calibrated by setting the maximum of the U M$_4$ absorption white line of UO$_2$ reference sample to 3725 eV. The energy was scanned between 3721 eV and 3736 eV for UX$_4$ compounds and between 3718 eV and 3736 eV for UX$_3$ compounds. XANES spectra were measured in HERFD mode by recording the intensity at the maximum of the U Mβ emission line (4f$_{5/2}$ → 3d$_{3/2}$) as a function of the incident energy, with a total energy resolution of ~0.7 eV. The RIXS data was measured by scanning the incident energy at different emission energies around the Mβ emission line, near the U M$_4$ edge. The emission energy was selected by an X-ray emission spectrometer equipped with five spherically bent Si(220) crystal analyzers with a 1.0 m bending radius at a Bragg angle of 75° in a vertical Rowland geometry[20]. The detected intensity was recorded by a Ketek detector with a 10 mm diameter and 450 mm Si thickness. Radiation tests were performed on each sample in the UX$_3$ and UX$_4$ series by

repeated counts of 0.1 s to check if any variation of the intensity took place.

### RIXS and HERFD calculations

The 3d-4f RIXS maps were calculated using multiplet calculations as implemented in the Quanty software library[54] starting from an input file generated using the Crispy graphical user interface[59]. The model included atomic and crystal field contributions. The values of the Slater-Condon integrals, i.e., the electrostatic (F$_k$) and exchange (G$_k$) interactions, and the spin-orbit coupling constants for the ground, intermediate, and final electronic configurations of the 3d-4f RIXS process are provided in Supplementary Table 1. The calculated RIXS maps were convoluted with a Lorentzian function to account for the lifetime broadening of the intermediate and final states, with a half-width at half-maximum (HWHM) of 3.54 eV and 0.28 eV, respectively. The maps were additionally broadened with a Gaussian function with HWHM of 0.6 eV to simulate the experimental response function. The HERFD spectra were subsequently extracted as cuts of the calculated RIXS maps along the incident energy axis at the constant emitted energy at the maximum of the RIXS intensity. The crystal field term was added to include the influence of local coordination and ligand type on the spectral changes. Details related to the process of extracting the required crystal field parameters are presented in the Supplementary Information.

### Inclusion of crystal field contributions in the multiplet calculations

To estimate the effects of local coordination and orbital energetics on the 5f orbitals of U$^{IV}$ and U$^{III}$ halides, density functional theory calculations (DFT) were performed for all UX$_4$ and UX$_3$ halides considered in the manuscript. Self-consistent calculations were carried out using the Full Potential Local-Orbital code (FPLO)[55], with the PBE density functional in a scalar relativistic approach. The Brillouin zones were sampled using a regular 12 x 12 x 12 k-point mesh.

Electronic band structures were calculated for all halide systems and projected on localized Wannier orbitals (see Supplementary Fig. 11a). The local basis set included U 5f, 6d, 7s, and the halide p orbitals (2p, 3p, 4p, and 5p for F, Cl, Br, and I, respectively). Following the Wannierization process, we extracted a numerical representation of the crystal field Hamiltonian on the basis of symmetric function (Supplementary Fig. 11b). The values reflect the influence of the local

environment on the 5f orbitals (symmetry, ligand type, and bond distances). Finally, we performed a one-to-one mapping of the numerical Hamiltonian with an analytical version we obtained from the Quanty documentation to extract the parameters that describe the crystal field in different point group symmetries.

The crystal field parameters obtained from the Wannierization process were used to calculate the 3d-4f RIXS maps using the semi-empirical multiplet theory as implemented in the Quanty software library[54]. The 3d-4f RIXS spectra were calculated considering the actual point group symmetries of the compounds: C2 for $UF_4$, $UBr_4$, and $UI_4$, D4h for $UCl_4$, and C3 for $UBr_3$ and $UCl_3$.

## Quantum theory of atoms in molecules (QTAIM) calculations

The Quantum Theory of Atoms in Molecules (QTAIM) metrics, $\rho$ and $\nabla^2\rho$ were calculated using the Critic2[60] code using the electron density calculated using the ORCA package[61]. The structural models required by the ORCA calculation were extracted from the crystallographic structure of the compounds. The models include the uranium atom and the atoms from the first coordination shell. Extensive tests using larger clusters showed that QTAIM metrics are insensitive to the presence of atoms that extend beyond the halogens directly bound to uranium. Technical details related to the ORCA and Critic2 calculations can be found in the Supplementary Information.

## Synthesis of $UX_4$ (X = F, Cl, Br, I)

All operations with $UF_6$ were performed in either stainless steel (316 L) or Monel metal Schlenk lines, which were passivated with 100% fluorine at various pressures before use. Preparations were carried out in an atmosphere of dry and purified argon (5.0, Praxair) using high-vacuum glass lines or a glovebox (MBraun), if not stated differently. Hydrogen fluoride (99%, Hoechst) was dried over $K_2NiF_6$ prior to its use. The well-educated reader is aware that $F_2$, HF, and $UF_6$ can be dangerous if not properly handled. Uranium compounds are radioactive and should therefore not be ingested. Sulfur is safe at ambient conditions.

**Preparation of $UF_4$[62].** In a typical reaction, sublimed $S_8$ (455 mg, 14.2 mmol) was placed in a perfluorinated ethylene-propylene copolymer (FEP) reaction tube and heated in vacuo several times. Approximately 2 mL of anhydrous HF was added by vacuum distillation. The suspension was frozen with liquid nitrogen and $UF_6$ (3.27 g, 9.29 mmol) was distilled onto it. The reaction mixture was slowly warmed, and the reaction began at the melting point of HF. The supernatant solution turned bluish for a few seconds, then brown with the formation of a greyish solid. After two to three days of reaction time at room temperature, a green solid formed, with a colorless supernatant HF solution. The solvent and volatile reaction products were distilled into a separate FEP reaction tube and the crude product was transferred into a flame-dried glass Schlenk tube. The tube was attached to a vacuum line and residual $S_8$ was sublimed off in a fine vacuum at approximately 350 °C. The yield of $UF_4$ is quantitative with respect to $UF_6$.

**Preparation of $UCl_4$, $UBr_4$ and $UI_4$[63].** Aluminum chloride and bromide (Merck, 98%/ Alfa Aesar, 98%) were purified by sublimation *in vacuo*. Iodine was sublimed *in vacuo* twice, the first time from phosphorous pentoxide. Aluminum (Fluka, purum >99%), as well as uranyl nitrate (Riedel de Haën, zur Analyze), were used as supplied. All glass vessels were made of borosilicate glass and flame-dried under vacuum before use. For the syntheses, glass ampoules were used as shown in Supplementary Fig. 13.

**Preparation of $UO_2$.** 26.2 g of $UO_2(NO_3)_2\cdot6H_2O$ (52 mmol) were decomposed to 14.5 g $U_3O_8$ (17 mmol) by heating to 700 °C in air for 7 h inside an open fused silica test tube. The black product was powdered in air and reduced in a stream of hydrogen at 800 °C for 8 h to obtain 13.8 g (51 mmol, 98 %) of phase pure $UO_2$.

**Preparation of $UCl_4$.** An ampoule was charged with 1.080 g $UO_2$ (4 mmol) and 1.066.7 + 0.177 g $AlCl_3$ (8 mmol + transport agent) and flame sealed under vacuum ($1\cdot10^{-3}$ mbar). The starting products were reacted at 250 °C for 5 h before the transport reaction was conducted with a source temperature of 350 °C and a sink temperature of 250 °C. 1.476 g (3.9 mmol, 97 %) of dark green crystals of $UCl_4$ were extracted after three days of transport.

**Preparation of $UBr_4$.** An ampoule was charged with 544 mg $UO_2$ (2 mmol) and 1075 + 65 mg $AlBr_3$ (4 mmol + transport agent) and flame sealed under vacuum ($1\cdot10^{-3}$ mbar). The starting products were reacted at 250 °C for 12 h before the transport reaction was conducted with a source temperature of 350 °C and a sink temperature of 230 °C. 991 mg (2.2 mmol, 88 %) of large brown plate-shaped crystals were extracted after 6 days.

**Preparation of $UI_4$.** An ampoule was charged with 270 mg $UO_2$ (1 mmol), 65 mg Al (2.4 mmol), and 2670 mg $I_2$ (10.5 mmol) and flame sealed under vacuum ($1\cdot10^{-3}$ mbar). The starting products were reacted at 150 °C for 1 h to prevent a too vigorous reaction of Al and $I_2$ followed by 350 °C for 6 h before the transport reaction was conducted with a source temperature of 450 °C and a sink temperature of 300 °C. 607 mg (0.8 mmol, 81 %) of black lath-like crystals were extracted after eight days.

## Synthesis of $UX_3$ (X = Cl, Br, I)[64]

For the synthesis of $UCl_3$, $UBr_3$, and $UI_3$, ampoules similar to those used for the synthesis of the $U^{IV}$ halides were used. The size should be such that a consistent temperature is achieved over the whole length during the reaction and that after the reaction a temperature gradient can be applied for the chemical vapor transport. Ampoules of about 100 mm in length, 19 mm outer diameter, and 1.3 mm wall thickness suit our tubular furnaces. These ampoules are equipped with a NS14 ground joint for flame drying and sealing. A constriction in the middle of the ampoule is used for the facilitated breakup of the ampoule and prevents the mixing of byproducts in the sink with products in the source. The ampoules are filled with a long funnel in a glovebox, to prevent soiling the sink as well as the sealing constraint with non-volatile substances.

**Preparation of $UCl_3$.** An ampoule was charged with 1000 mg of finely ground $UCl_4$ (2.633 mmol, 20 mg excess) and 18.12 mg Si (0.645 mmol) and flame sealed under vacuum ($1\cdot10^{-3}$ mbar). The starting materials were reacted at 450 °C for 20 days before the transport reaction was conducted with a source temperature of 350 °C and a sink temperature of 250 °C, to remove the excess $UCl_4$. The yield is quantitative with respect to silicon.

**Preparation of $UBr_3$.** An ampoule was charged with 1000 mg of finely ground $UBr_4$ (1.793 mmol, 20 mg excess) and 12.34 mg Si (0.440 mmol) and flame sealed under vacuum ($1\cdot10^{-3}$ mbar). The starting materials were reacted at 400 °C for 14 days before the transport reaction was conducted with a source temperature of 350 °C and a sink temperature of 230 °C, to remove the excess $UBr_4$. The yield is quantitative with respect to silicon.

**Preparation of $UI_3$.** An ampoule was charged with 1000 mg of finely ground $UI_4$ (1.341 mmol, 20 mg excess) and 9.23 mg Si (0.329 mmol) and flame sealed under vacuum ($1\cdot10^{-3}$ mbar). The starting materials were reacted at 450 °C for 7 days before the transport reaction was conducted with a source temperature of 450 °C and a sink temperature of 300 °C. The yield is quantitative with respect to silicon.

## Data availability

The data generated in this study have been deposited in the Zenodo database under accession code 10.5281/zenodo.12706853.

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

## Acknowledgements

The authors would like to acknowledge the Rossendorf Beamline of the ESRF for providing beamtime and J. Exner, D. Naudet, and N. Baumann for their computational and technical support during the experiments. This research was funded by the European Research Council (ERC) under grant agreement No. 759696 (K.O.K.).

## Author contributions

The work was conceived and designed by K.O.K.; F.K. and T.G. synthesized samples at Philipps-Universität Marburg; S.W. sealed synthesized materials at HZDR under anoxic conditions and transported them to the beamline at ESRF; K.O.K. and E.F.B. performed HERFD and RIXS measurements at ROBL of ESRF; C.L.S. carried out the theoretical calculations and analyzed the experimental data together with L.A. and M.R.; C.L.S., L.A., and K.O.K. wrote the manuscript with input from all authors.

## Funding

## Competing interests

The authors declare no competing interests.
