## [Peer Review File · Nature Communications]

On the origin of low-valent uranium oxidation stateReviewer #1 (Remarks to the Author):

Review on Nature Communications Manuscript NCOMMS-23-59442A-Z

On the origin of low-valent uranium oxidation state

by C. L. Silva, L. Amidani, M. Retegan, E. F. Bazarkina, S. Weiss, T. Graubner, F. Kraus, and K. O. Kvashnina

C. L. Silva et al. present an excellent and timely U M_4 HERFD-XANES / 3d4f-RIXS study, focusing on tri- and tetravalent U halides $U^{III}X_3$ and $U^{IV}X_4$, respectively, with X = F, Cl, Br, I. The nature of the electronic bond in 5f-element compounds is a persisting hot and sometimes controversially discussed topic in nuclear chemistry, driven by rapidly expanding experimental capabilities (namely advanced X-ray spectroscopies) and supported by more and more efficient electronic structure calculations. So far, the focus has been on the higher oxidation states of the actinide elements, which are generally easier to stabilize. In the present study, a systematic comparison between U^{III} and U^{IV} systems has been achieved, where trivalent U species have been for the first time ever investigated by M-edge HERFD methods. The different trends and effects in the tri- and tetravalent halide series are concisely described and can be well reproduced in the framework of crystal field multiplet theory. Beam induced effects, which might explain the spectral trends at the first glimpse, are discussed and convincingly ruled out. Additional material in the SI is adequately used to underpin the findings and their interpretation.

Just a few suggestions:

- line 50: a "." missing after the (X=...) brackets.
- line 134: I assume it should be "turned on and off"
- line 164: the same, "turned off"
- line 182-183 - better: ... compared to the theoretical one obtained by cutting the calculated RIXS of Figures 3c and 3d, respectively.
- line 252-254 – better: Therefore, one expects an increased localization on the halide ligand set as the metal oxidation state is lowered from ...
- line 315 ff.: How was the incident beam / excitation energy scale calibrated?

Overall recommendation: publish the manuscript and SI in the present form with only minor modifications as listed above.

Review for manuscript from Kvashnina et al “On the origin of low-valent uranium oxidation state”

The manuscript reports experimental and theoretical results of a systematic study on the effects of ligand and oxidation state on the series UX_3 and UX_4 for $X = F, Cl, Br,$ and I . Systematic studies like this one are essential for gaining deeper insights on the nature of chemical bonding and the influence of ligand and metal oxidation state on the spectroscopic features observed in x-ray spectroscopy, including HERFD, RIXS, and other techniques. So, I applaud the authors on the intellectual design of the experiment.

The experimental and computational studies are expertly performed by this group, and the insights from this kind of systematic study are extremely valuable to the scientific community. The uranium M_4 HERFD experiments reveal useful white line shifts that are sensitive to oxidation state, and the high resolution of the HERFD helps to quantify the magnitude of this shift between U(III) and U(IV) in this study. These results are then compared to prior published work on U(V) and U(VI), providing a good understanding of the applicability of HERFD for oxidation state characterization of uranium between oxidation states III – VI.

The authors do a good job of simulating the RIXS maps through electronic structure theory, indicating that one can glean some theoretical understanding of the origin of spectroscopic features observed. I recommend publication, but ask for some revisions in the text related to clarity of presentation.

I felt that that authors skipped some important background on how these RIXS experiments actually work, what we're actually measuring, and limitations of the computational approach. They know the details, because this group has published expert reviews on these techniques in the past. However, if the reader is not already an expert, it is not clear from the discussion in many cases, exactly what is being measured, leading to some confusion. This can be readily addressed in a modification to the text.

There are also a few issues the reviewer has with some of the discussion used in the manuscript, and I recommend some clarification.

- 1) On the third page the authors comment that the two-peak structure in the U(III) HERFD spectra indicates an impurity in the compounds, owing to the instability of the U(III) oxidation state. When I read the statement on page three “one can assume that a mixture of both III and IV oxidation states is present”, I almost didn't want to read the rest of the paper, because I thought the authors were performing sophisticated studies on impure mixtures of oxidation states! We later learn that the compounds are indeed pure, and that the two peak structure is a $U M_4$ spectroscopic signature for U(III). I recommend removing these impurity comments from the early portions of the manuscript and either provide clarity that ‘it will be

shown that the two-peak structure is NOT due to impurities”, or simply save this for the discussion section that appears later in the manuscript.

- 2) The authors have overlooked changes in coordination number as part of their discussion surrounding the UX_4 series in Table 2, and the ensuing interpretation of the spectroscopy. The text goes into a lengthy and somewhat naïve discussion that claims that the changes in the M4 HERFD XANES spectra are due to electronegativity differences across the halide series. There is a nonsensical statement below Table 2 that states “Element with significantly different electronegativities are more likely to form ionic bonds”. They completely ignore the fact that the coordination number changes from 8 to 7 to 6 upon progressing from UCl_4 to UBr_4 to UI_4 . We know that orbital energetics and molecular symmetry (coordination number) play a significant role in the electronic structure. For bonding, the halide frontier orbitals and radial functions change from 2p, 3p, 4p and 5p, with concomitant changes in energy of the lone pairs as we move from F to I. The focus on electronegativity and ionic nature of bonding misses this point altogether. I note that UF_6 is well-known to be quite covalent in its U-F bonding, going against the electronegativity arguments presented here, because orbital energies and overlap functions matter. We know it’s a combination of orbital energetics between F 2p and U 5f and 6d orbitals that matter for UF_6 bonding. Surely the orbital energetics of the np orbital is important in the UX_4 and UX_3 series as well. This entire discussion needs to be reconsidered prior to publication.
- 3) A computational experiment that would address the above concern would be to perform calculations on UF_4 in the known geometries for UBr_4 and UI_4 which means for coordination numbers 7 and 6. For identical F ligands, one can then calculate whether spectroscopic changes observed in HERFD spectra are due to coordination number change from 8 to 7 to 6 using the same ligand. This would be more convincing than a hand waving argument, and would either point to the role of coordination number/geometry, or rule it out.
- 4) It would be valuable to explain to the reader how atomic multiplet calculations that focus on atomic-like transitions give insights into covalency where electrons are ‘not’ atomic-like, but delocalized. Chemical bonding takes place in lower lying orbitals. Ligand-based orbitals overlap with actinide 5f or 6d orbitals to form bonding (ligand-based) and antibonding (metal-based) molecular orbitals. So the electrons involved in covalent chemical bonding are supplied by the ligands, not the metal. Therefore the bonding electrons are not 5f or 6d electrons. Statements that say we’re exploring the role of 5f electrons in chemical bonding are misleading. This is readily addressed by modification of the discussion in the text.

Response to Reviewers Reports

Manuscript ID: NCOMMS-23-59442A-Z

Title: "On the origin of low-valent uranium oxidation state"

Author(s): C. L. Silva, L. Amidani, M. Retegan, E. F. Bazarkina, S. Weiss, T. Graubner, F. Kraus, and K. O. Kvashnina

Dear Reviewers,

Thank you for reviewing our manuscript 23-59442A-Z and for the valuable feedback. We have carefully improved the manuscript, based on your comments.

We would like to thank particularly reviewer 2. His/her valuable comments stimulated us to review part of the discussion and perform additional calculations as detailed below.

To investigate the effect of the coordination number, bond distances, and different halide ligands on the spectroscopic features, we recalculated RIXS and HERFD-XANES by implementing an accurate crystal field (CF) model. The parameters for the model were obtained by:

- Performing band structure calculations using Density Functional Theory (DFT) as implemented in the FPLO code.
- Projecting the band structure onto localised Wannier orbitals using FPLO.
- Transforming the tight-binding Hamiltonian to extract the CF parameters as outlined in the Quany documentation.

To understand the spectral changes in the set of U^{IV} halides beyond the simplistic explanation based on the ligand's electronegativity, we performed quantum theory of atoms in molecules (QTAIM) calculations to determine if covalency changes could explain the trends observed. QTAIM bond critical point analysis with Critic2 code indeed finds a trend in decreased ionicity from UI_4 to UF_4 , in agreement with the trends in our spectra, as predicted by reviewer 2.

Altogether, the manuscript has been enriched by an additional theoretical analysis and accurate evaluation of the effects of the local environment to rationalize the trends of the UX_4 and UX_3 series.

In the following pages, we respond to the comments point-by-point and identify changes to the main text. We hope that these additions provide greater context and clarity to our findings.

Sincerely, Kristina Kvashnina on behalf of all authors

Reviewer(s)' Comments to Author:

Reviewer: 1

Comments:

C. L. Silva et al. present an excellent and timely U M₄ HERFD-XANES / 3d4f-RIXS study, focusing on tri and tetravalent U halides U^{III}X₃ and U^{IV}X₄, respectively, with X = F, Cl, Br, I. The nature of the electronic bond in 5f-element compounds is a persisting hot and sometimes controversially discussed topic in nuclear chemistry, driven by rapidly expanding experimental capabilities (namely advanced X-ray spectroscopies) and supported by more and more efficient electronic structure calculations. So far, the focus has been on the higher oxidation states of the actinide elements, which are generally easier to stabilize. In the present study, a systematic comparison between U^{III} and U^{IV} systems has been achieved, where trivalent U species have been for the first time ever investigated by M-edge HERFD methods. The different trends and effects in the tri- and tetravalent halide series are concisely described and can be well reproduced in the framework of crystal field multiplet theory. Beam induced effects, which might explain the spectral trends at the first glimpse, are discussed and convincingly ruled out. Additional material in the SI is adequately used to underpin the findings and their interpretation.

Overall recommendation: publish the manuscript and SI in the present form with only minor modifications.

We thank the reviewer for the positive feedback. We have improved the manuscript according to the suggestions, as detailed below.

- Line 50: a “.” missing after the (X=...) brackets.

The “.” was added.

- Line 134: I assume it should be “turned on and off”

Corrected.

- Line 164: the same, “turned off”

Corrected.

- Line 182-183 - better: ... compared to the theoretical one obtained by cutting the calculated RIXS of Figures 3c and 3d, respectively.

This statement has been corrected.

- Line 252-254 – better: Therefore, one expects an increased localization on the halide ligand set as the metal oxidation state is lowered from ...

This statement has been corrected.

- Line 315 ff.: How was the incident beam / excitation energy scale calibrated?

The incident beam was calibrated by setting the maximum of the U M₄ white line of the UO₂ reference to 3725 eV. We added this information in the experimental section of the methods.

Reviewer: 2

Comments:

The manuscript reports experimental and theoretical results of a systematic study on the effects of ligand and oxidation state on the series UX₃ and UX₄ for X = F, Cl, Br, and I. Systematic studies like this one are essential for gaining deeper insights on the nature of chemical bonding and the influence of ligand and metal oxidation state on the spectroscopic features observed in x-ray spectroscopy, including HERFD, RIXS, and other techniques. So, I applaud the authors on the intellectual design of the experiment.

The experimental and computational studies are expertly performed by this group, and the insights from this kind of systematic study are extremely valuable to the scientific community. The uranium M₄ HERFD experiments reveal useful white line shifts that are sensitive to oxidation state, and the high resolution of the HERFD helps to quantify the magnitude of this shift between U(III) and U(IV) in this study. These results are then compared to prior published work on U(V) and U(VI), providing a good understanding of the applicability of HERFD for oxidation state characterization of uranium between oxidation states III – VI.

The authors do a good job of simulating the RIXS maps through electronic structure theory, indicating that one can glean some theoretical understanding of the origin of spectroscopic features observed. I recommend publication, but ask for some revisions in the text related to clarity of presentation.

We thank the reviewer for the detailed and very thoughtful review.

I felt that that authors skipped some important background on how these RIXS experiments actually work, what we're actually measuring, and limitations of the computational approach. They know the details, because this group has published expert reviews on these techniques in the past. However, if the reader is not already an expert, it is not clear from the discussion in many cases, exactly what is being measured, leading to some confusion. This can be readily addressed in a modification to the text.

We agree that this information is missing and should be added for non-expert readers. Additional information about RIXS and HERFD experiments as well as computational approaches has been added. A schematic of the RIXS process has also been included (see Figure 1a). In the results and methodology sections, we improved the description of the process to clarify what is being measured. Additionally, we improved the description of the calculation in the results section and addressed the limitations of the computational approach.

There are also a few issues the reviewer has with some of the discussion used in the manuscript, and I recommend some clarification.

1) On the third page the authors comment that the two-peak structure in the U(III) HERFD spectra indicates an impurity in the compounds, owing to the instability of the U(III) oxidation state. When I read the statement on page three “one can assume that a mixture of both III and IV oxidation states is present”, I almost didn't want to read the rest of the paper, because I thought the authors were performing sophisticated studies on impure mixtures of oxidation states! We later learn that the compounds are indeed pure, and that the two peak structure is a U M₄ spectroscopic signature for U(III). I recommend removing these impurity comments from the early portions of the manuscript and either provide clarity that “it will be shown that the two-peak structure is NOT due to impurities”, or simply save this for the discussion section that appears later in the manuscript.

We thank the reviewer for pointing this out. We removed the statement about the impurity at the beginning of the results section and discussed it differently.

2) The authors have overlooked changes in coordination number as part of their discussion surrounding the UX₄ series in Table 2, and the ensuing interpretation of the spectroscopy. The text goes into a lengthy and somewhat naïve discussion that claims that the changes in the M₄ HERFD XANES spectra are due to electronegativity differences across the halide series. There is a nonsensical statement below Table 2 that states “Element with significantly different electronegativities are more likely to form ionic bonds”. They completely ignore the fact that the coordination number changes from 8 to 7 to 6 upon progressing from UCl₄ to UBr₄ to UI₄. We know that orbital energetics and molecular symmetry (coordination number) play a significant role in the electronic structure. For bonding, the halide frontier orbitals and radial functions change from 2p, 3p, 4p and 5p, with concomitant changes in energy of the lone pairs as we move from F to I. The focus on electronegativity and ionic nature of bonding misses this point altogether. I note that UF₆ is well-known to be quite covalent in its U-F bonding, going against the electronegativity arguments presented here, because orbital energies and overlap functions matter. We know it’s a combination of orbital energetics between F 2p and U 5f and 6d orbitals that matter for UF₆ bonding. Surely the orbital energetics of the np orbital is important in the UX₄ and UX₃ series as well. This entire discussion needs to be reconsidered prior to publication.

We thank the reviewer very much for this valuable information, which stimulated us to estimate different impact of the local environment (coordination number, type of halide and bond distances) on M₄ spectra and guided us through a new set of calculations. Below we explain in details what have been added:

For the partially filled 5f shell of tri- and tetravalent U, the multiplet approach is the most adequate starting point. We, therefore, implemented an accurate crystal field (CF) model to include the effect of the local environment while keeping a correct treatment of electron-electron interactions. In the previous version of the manuscript, the effect of CF was tested with reasonable but arbitrary parameters. In light of the reviewer's comment, we extracted the CF parameters from *ab initio* calculation using the proper local symmetry of uranium. The parameters were then used to run new RIXS and HERFD calculations.

First, we performed DFT calculations as implemented in the FPLO code¹ on all UX₄ and UX₃ halides. The band structures were projected on localized Wannier orbitals (see Report Fig. 1, panel a). The local basis set included U 5f, 6d, 7s and the halogen p orbitals. From the Wannierization process,² we obtained the eigenvalues of the 5f orbitals. In a CF model, the influence of the local environment on the 5f orbitals (coordination, type of ligand, bond distances) causes the change of the local symmetry and the lifting of the 5f degeneracy. The magnitude of the 5f splitting reflects the strength of the interaction with the ligands. The splitting of the 5f obtained from the Wannierization is shown in Report Fig. 1d. We extracted the crystal field parameters corresponding to the 5f splitting (see Report Fig. 1, panels b and c) and ran new RIXS and HERFD calculations, adding the implemented crystal field to the atomic Hamiltonian (see Report Fig. 1, panel d).

Report Fig 1. **Wannierization process.** (a) Electronic band structure calculations. The background (in black) is the electronic band structure calculated by FPLO. The projected Wannier bands are shown in different colours according to each orbital (blue for U 6d, red for U 5f, pink for Cl 3p). (b) Numerical crystal field Hamiltonian matrix obtained following wannierization. (c) Analytical crystal field Hamiltonian matrix. The eigenstates (diagonal elements) and mixing parameters (off-diagonal elements) in cubic harmonics were extracted from the matrix in (c). (d) 5f splitting of UX_4 systems due to the implemented crystal field.

The results of the comparison between crystal field and atomic calculations were added to the new manuscript. Report Fig. 2 shows one example, the case of UCl_4 . Compared to purely atomic calculations, the crystal field slightly reduces the second post-edge feature.

Report Fig. 3 summarizes the work that has been done to understand the U M_4 HERFD-XANES data on the UX_4 series. Panel (a) shows the results for the atomic model with reduction of the Slater-Condon integrals by 15 % from UI_4 to UF_4 . Panel (b) shows results for the combination of the crystal field model including the local geometry (coordination, point symmetry, bond distances) and the modulation of the Slater-Condon integrals. The reduction factors are similar to those of the atomic calculations. Panel (c) shows the results for the crystal field model without modulation of the Slater-Condon integrals. The differences shown at the bottom indicate that the changes in panels (a) and (b) are in the direction of the experimental changes. This analysis shows that the intra-atomic interactions are the principal force shaping the U M_4 edge data for these materials. Including the modulation of the Slater-Condon integrals in the CF model reproduces the experimental changes very well. These results have been added to Fig. 4 and Supplementary Fig. 3-6.

Report Fig 2. **Comparison between atomic and crystal field HERFD-XANES calculations at the U^{IV} M_4 -edge for UCl_4 .**

Report Fig 3. **Investigation of the optimal model to reproduce the HERFD-XANES data at U M₄ edge for the U(IV) halide systems.** (a) Atomic model with 15 % reduction in Slater-Condon integrals from UI₄ to UF₄. (b) Crystal field model considering the local geometry around the absorbing atom. The Slater-Condon integrals are modulated on top of the accurate crystal field model with reduction factors similar to those in panel a. (c) Crystal field model without modulation of the Slater-Condon integrals.

In addition, to understand the trends observed in the UX₄ series, we abandoned the simplistic electronegativity argument, as suggested by Reviewer 2, and performed Quantum Theory for Atoms in Molecules (QTAIM) analysis to estimate the ionicity/covalency of the U-halogen bond. The method has been used previously to analyze the bonding in the actinides series.³

We used the values of the magnitude of the electron density at the bond critical point (ρ_{BCP}) between the U atom and the halogen ligand to address the nature of the bonding. The results (Report Table 1) show that the halide bonds are characterized by ionic interactions ($\rho_{\text{BCP}} < 0.1$ a.u.), with the degree of covalent character increasing from UI₄ ($\rho_{\text{BCP}} = 0.048$ a.u.) to UF₄ ($\rho_{\text{BCP}} = 0.076$ a.u.). In light of these new results, we dropped the electronegativity argument as suggested by the reviewer and we discuss trends in UX₄ and UX₃ series based on QTAIM.

Report Table 1. **QTAIM results.** Bond critical point (ρ_{BCP})

System	ρ_{BCP} (a.u.)	$\nabla^2\rho$
UF ₄	0.076	0.293
UCl ₄	0.053	0.114
UBr ₄	0.052	0.092
UI ₄	0.048	0.061

3) A computational experiment that would address the above concern would be to perform calculations on UF₄ in the known geometries for UBr₄ and UI₄ which means for coordination numbers 7 and 6. For identical F ligands, one can then calculate whether spectroscopic changes observed in HERFD spectra are due to coordination number change from 8 to 7 to 6 using the same ligand. This would be more convincing than a hand waving argument, and would either point to the role of coordination number/geometry, or rule it out.

We thank the reviewer for the suggestion and for the opportunity to improve our theoretical analysis. The results are shown in Report Figure 4. The same method described in the previous answer (DFT + Wannierization to obtain CF parameters) was applied to the UCl₄ and UI₄ structure where Cl and I were substituted with F. We obtained in this way U coordinated with 8 and 7 F. The self-consistent field for the

DFT calculation of F in UI_4 did not converge; therefore only 3 compounds are shown here. The results show that the spectra for F in the structure of UF_4 , UCl_4 and UBr_4 are almost identical, which rules out the relevance of the coordination number as relevant parameter. Results of this computational experiment has been added as Supplementary Fig. 7.

Report Fig 4. **Investigation of the effects of the coordination number (CN) and local geometry around the absorbing atom.** The calculations were performed for UCl_4 and UBr_4 with Cl and Br replaced by F.

4) It would be valuable to explain to the reader how atomic multiplet calculations that focus on atomic-like transitions give insights into covalency where electrons are ‘not’ atomic-like, but delocalized. Chemical bonding takes place in lower lying orbitals. Ligand-based orbitals overlap with actinide 5f or 6d orbitals to form bonding (ligand-based) and antibonding (metal-based) molecular orbitals. So the electrons involved in covalent chemical bonding are supplied by the ligands, not the metal. Therefore the bonding electrons are not 5f or 6d electrons. Statements that say we’re exploring the role of 5f electrons in chemical bonding are misleading. This is readily addressed by modification of the discussion in the text.

We agree with the reviewer. The M_4 edge HERFD-XANES probes the 5f electronic structure, which, especially for U(IV) and U(III), is not among the main actors of the bonding. Indeed, the direct way to estimate covalency is to probe the ligand K-edge, where the mixing of 5f can be directly seen features related to the metal in the ligand p-projected electronic structure. However, the formation of the bond also affects the 5f electrons if the 5f orbitals are partially involved in the bond. How this mixing shows up and can be quantified from $M_{4,5}$ edges is not yet well understood and is at the center of current research. With this manuscript we bring new evidences to this discussion. We find that to reproduce the experimental trends it is necessary to reduce the interactions of localized 5f electrons going from I to F. This lowering is an effect of increased delocalization of 5f orbitals, therefore correlated with covalency. This interpretation is supported by QTAIM results, which find a similar trend for covalency. We modified the text to make these points clear to the reader.

REFERENCES

1. Koepernik, K. & Eschrig, H. Full-potential nonorthogonal local-orbital minimum-basis band-structure scheme. *Phys. Rev. B* 59, 1743–1757 (1999).
2. Haverkort, M. W., Zwierzycki, M. & Andersen, O. K. Multiplet ligand-field theory using Wannier orbitals. *Physical Review B - Condensed Matter and Materials Physics* 85, (2012).
3. Tanti, J., Lincoln, M. & Kerridge, A. Decomposition of d- and f-shell contributions to uranium bonding from the quantum theory of atoms in molecules: Application to uranium and uranyl halides. *Inorganics* 6, (2018).

REVIEWERS' COMMENTS

Reviewer #2 (Remarks to the Author):

The revised manuscript is excellent. I thank the authors for acting on my comments, and performing more detailed analyses. In the end, this is a much better contribution to the scientific community, and the deeper background in the introduction will be very helpful to all the readers and their students.

Let me also say that I gained a deeper understanding of the techniques from my review and study of this paper. thank you for letting me review it!

Please publish the revised manuscript as is.

I recognize that Nature Communications has a transparent review approach. However, I'm still rather old-school, and believe that anonymous review strengthens our field, and is the key to being able to engage in meaningful debate during the review process.